# Role of Magnesium in the Intensive Care Unit and Immunomodulation: A Literature Review

**DOI:** 10.3390/vaccines11061122

**Published:** 2023-06-20

**Authors:** Francesco Saglietti, Alessandro Girombelli, Stefano Marelli, Francesco Vetrone, Mario G. Balzanelli, Payam Tabaee Damavandi

**Affiliations:** 1Santa Croce and Carle Hospital, Department of Emergency and Critical Care, 12100 Cuneo, Italy; 2Division of Anesthesiology, Department of Anesthesiology, Intensive care and Emergency Medicine, Ospedale Regionale di Lugano, 69000 Lugano, Switzerland; alessandro.girombelli91@gmail.com; 3Department of Medicine and Surgery, University of Milan-Bicocca, 20900 Monza, Italy; s.marelli94@gmail.com (S.M.); francesco.vetrone@asst-nordmilano.it (F.V.); 4Department of Prehospital Emergency Medicine, ASL TA, Italian Society of Prehospital Emergency Medicine (SIS 118), 74121 Taranto, Italy; mario.balzanelli@gmail.com; 5Department of Neurology, Fondazione IRCCS San Gerardo dei Tintori, School of Medicine and Surgery, Milan Center for Neuroscience, University of Milano-Bicocca, 20900 Monza, Italy; p.tabaeedamavandi@campus.unimib.it

**Keywords:** magnesium, critical care, immunomodulation, infections

## Abstract

Both the role and the importance of magnesium in clinical practice have grown considerably in recent years. Emerging evidence suggests an association between loss of magnesium homeostasis and increased mortality in the critical care setting. The underlying mechanism is still unclear, but an increasing number of in vivo and in vitro studies on magnesium’s immunomodulating capabilities may shed some light on the matter. This review aims to discuss the evidence behind magnesium homeostasis in critically ill patients, and its link with intensive care unit mortality via a likely magnesium-induced dysregulation of the immune response. The underlying pathogenetic mechanisms, and their implications for clinical outcomes, are discussed. The available evidence strongly supports the crucial role of magnesium in immune system regulation and inflammatory response. The loss of magnesium homeostasis has been associated with an elevated risk of bacterial infections, exacerbated sepsis progression, and detrimental effects on the cardiac, respiratory, neurological, and renal systems, ultimately leading to increased mortality. However, magnesium supplementation has been shown to be beneficial in these conditions, highlighting the importance of maintaining adequate magnesium levels in the intensive care setting.

## 1. Introduction

Magnesium is an essential element within the human body, and takes part in many biochemical reactions. The potential clinical importance of this ion began to emerge as early as the 1960s [1]. This metal interacts with the human body both as an ion, and as a fundamental cofactor in numerous enzymatic reactions that regulate metabolism and protein synthesis, and maintain cellular integrity [2,3]. For years, the intricate relationship between magnesium and the immune system has been studied. A substantial body of evidence supports magnesium’s interaction with, and regulation of, the immune system, encompassing both nonspecific and specific immune responses. These findings have predominantly emerged from studies employing animal models, with a primary emphasis on investigating the consequences of a magnesium-depleted diet on animals [4].

While there are still unanswered questions regarding the precise immunological and molecular mechanisms by which magnesium acts, emerging evidence has shed light on its impact within clinical settings.

Both hypermagnesemia [5] and hypomagnesemia [6] are known to induce physiological alterations that contribute to the development of various diseases [7]. Moreover, the therapeutic potential of magnesium sulfate (MgSO_4_) has been explored in critical care conditions, such as ischemic and hemorrhagic stroke [8,9], head trauma [10], arrhythmias [11,12], asthmatic status [13,14], and preeclampsia [15]. A significant role is played by MgSO_4_ as an adjunct in both general and regional anesthesia. It is a relatively safe drug, with a fairly wide therapeutic range. Severe symptoms of intoxication appear when serum concentrations exceed five times the normal value [16,17]. A preinduction dose of 60 mg/kg of MgSO_4_ results in the significant reduction in the nondepolarizing neuromuscular blocking agent necessary for orotracheal intubation, without any notable side effects or increased need for reversal agents [18]. Guler et al. [19] demonstrated that pretreatment with 2.48 mmol intravenous MgSO_4_ reduced the incidence and intensity of etomidate-induced myoclonic movements during anesthesia induction, thereby dampening one of etomidate’s side effects. Magnesium has also been incorporated into opioid-free and opioid-sparing anesthesia protocols, to minimize or potentially eliminate the need for opioids. Di Benedetto et al. [20] reported reduced postoperative pain within the first 24 h after surgery, decreased use of rescue analgesic drugs, and a lower incidence of postoperative nausea and vomiting, using an opioid-free approach based on the concomitant administration of ketamine and magnesium. Additionally, MgSO_4_ is recognized as an important adjunct in regional anesthesia, as it accelerates the onset time of perineurally administered local anesthetics [21], prolongs the duration of the block, and decreases the numeric rating scale (NRS) scores at 6 and 12 h, resulting in a reduced use of rescue analgesics [22].

Based on the literature available, it can be reasonably inferred that magnesium serves as an essential adjunct in anesthesia practices, with potential implications for patient outcomes in the intensive care unit (ICU), including overall clinical prognosis, and treatment effectiveness. Surprisingly, to date, there are no reviews that thoroughly explain the interplay between magnesium, its immunomodulatory activity, and the clinical implications in the ICU setting. The aim of this review is to provide an overview of the available evidence regarding the role of magnesium in immune-dysregulated pathologies in the critical care setting, with specific attention given to its impact on mortality outcomes.

## 2. Materials and Methods

We performed a literature search using MEDLINE (accessed by Pubmed), the Cochrane Central Register of Controlled Trials (CENTRAL), and the US National Institutes of Health Clinical Trials Registry (http://www.clinicaltrials.gov (accessed on 16 March 2023)) from inception to week three of March 2023. The search strategy included “Magnesium”, “Sepsis”, “Immunomodulation”, “Intensive Care”, “ICU” and “Critical Care” as keywords in different combinations, using Boolean operators and Medical Subject Headings (MESH); no filters were applied. A total of 1083 articles were screened. Then, we excluded the following items: duplicates, papers different from original articles, non-English written papers, and any other publications that did not comply with the goal of the present review. Relevant articles were added through a search of the reference list. Overall, 109 articles were included in the present review.

## 3. Discussion

### 3.1. Magnesium and the Immune System

Magnesium, the second most abundant intracellular cation, plays a vital role in maintaining homeostasis within cells, and is crucial for the proper functioning of the immune system, specifically in regulating various types of immune cells [23,24]. Studies on animals have demonstrated that magnesium deficiency can lead to the activation of the innate immune system, and the impairment of the adaptive immune system, resulting in a pro-inflammatory state. In terms of the innate immune system, magnesium deficiency has been shown to activate polymorphonuclear leukocytes, leading to increased phagocytosis and oxidative stress [25]. Nonetheless, magnesium supplementation has been found to reduce monocyte cytokine production through a toll-like receptor (TLR) pathway [26]. Furthermore, magnesium plays a pivotal role in regulating acquired immunity, by influencing the development and proliferation of lymphocytes [27]. A reduced serum Mg^2+^ concentration in mice showed a marked reduction of CD8+ and CD4+ T-cell response against the influenza A virus [28]. Additionally, magnesium deficiency in mice has been associated with the early involution of the thymus, which negatively affects T-cell population [29]. The Mg^2+^ transporter TRPM7 is particularly important for T-cell development. When this transporter is absent, and thus magnesium supply is inadequate, T-cell development is inhibited, and cellular apoptosis is triggered. However, this effect can be partially reversed by culturing the cells in a medium enriched with high levels of magnesium [30]. In a mouse model with T-cell-specific deletion of the TRPM7 channel, the development of T lymphocytes was hindered at the CD4− CD8− stage, leading to reduced levels of CD4+ and CD4+ CD8+ cells in the thymus [31]. Extracellular magnesium has also been shown to regulate the effector function of CD8+ T-cells, via a co-stimulatory molecule LFA-1 mediated pathway [32]. In this context, multiple studies have demonstrated that even moderate or subclinical magnesium deficiency contributes significantly to chronic low-grade inflammation. The activation of leukocytes and macrophages, the release of inflammatory cytokines and acute-phase proteins, and the excessive production of free radicals are part of this inflammatory response [33]. A meta-analysis involving 32,198 individuals revealed a significant inverse association between dietary magnesium and serum C-reactive protein (CRP) levels, while magnesium supplementation seemed to reduce CRP levels [34]. The relationship between magnesium, inflammation, and the immune response has significant clinical implications, necessitating further investigation to enhance our understanding in this area.

### 3.2. Sepsis and Septic Shock

The association between serum magnesium and the pathophysiology of sepsis remains unclear. Animal studies have shown that hypomagnesemia is linked to elevated levels of IL6 and TNF-α, as well as increased activation of macrophages, neutrophils, and endothelial cells [4,35,36,37,38], while magnesium administration protects mice from lipopolysaccharide-induced lethal septic shock, by blocking gasdermin-D N-terminal-induced pyroptosis [39]. Hypomagnesemia may also hinder nitric oxide (NO) synthesis, thus increasing the risk of recurring infections, as NO plays a crucial role in preventing infections in body cavities, such as sinusitis, pneumonia, and mucositis [23,40], although reports on this matter are controversial [41]. Additionally, magnesium is a significant cofactor in the synthesis of thiamine pyrophosphate, which serves as a coenzyme in numerous enzymatic reactions. Inadequate levels of thiamine pyrophosphate can result in a reduced production of gastric acid, thereby increasing the risk of gastrointestinal (GI) infections [23]. These immune regulatory functions of magnesium may explain why hypomagnesemia has been linked to an increased recurrence of bacterial infections, such as sepsis, urinary tract infections, and bronchopneumonia [42].

Several studies have consistently reported a significant association between hypomagnesemia an increased incidence of sepsis or septic shock [43], prolonged ICU stay, and mortality in sepsis [44,45] (Table 1). For instance, a study by Thongprayoon et al. [46] on a group of 2589 patients with systemic inflammatory response syndrome (SIRS) without septic shock suggested that patients with SIRS and hypomagnesemia (defined as magnesium levels < 1.5 mg/dL) at the time of admission had a 1.86-fold greater risk of developing septic shock during their hospital stay.

In severe sepsis, serum lactate is an important indicator of tissue hypoperfusion, and a predictor of the sepsis outcome. Magnesium and thiamine are crucial cofactors in the Krebs cycle [47,48,49], and can impair adenosine triphosphate (ATP) production when they are lacking, leading to an increase in anaerobic metabolism, and the development of lactic acidosis [50]. This biological mechanism suggests that magnesium may play a role in increasing the rate of lactate clearance, by reducing lactate production or enhancing its elimination. To investigate the efficacy of magnesium supplementation in lactate clearance for patients with severe sepsis, Noormandi et al. enrolled 58 patients, and administered magnesium to maintain levels above 3 mg/dL The study demonstrated that magnesium supplementation increased lactate clearance from 37% to 47% on day 3 (*p* < 0.05) and reduced the time to lactate clearance (4.7 vs. 6.1 days). Patients in the magnesium group also had a significantly longer survival time than those in the placebo group (26 days versus 22 days, *p* < 0.01) [51].

Another important aspect of sepsis is coagulation abnormalities. Magnesium deficiency, as already underlined, promotes the secretion of inflammatory cytokines, and those cytokines lead to immune response dysregulation, and promote tissue damage, activating coagulation [52]. Tonai et al. analyzed the association between hypomagnesemia and septic coagulopathy, in a retrospective observational study involving 753 patients [53]. They were able to see an independent association of hypomagnesemia with disseminated intravascular coagulopathy (DIC) (OR 1.69, *p* = 0.048).

Considering the data mentioned above, it is not surprising that hypomagnesemia has been identified as an independent risk factor associated with inpatient mortality in a multivariable analysis [54]. This has been proven especially true in specific populations, such as elderly patients, patients with varying degrees of chronic kidney disease, and heart failure patients [55,56]. On the other hand, hypermagnesemia levels may have adverse effects on the pediatric population. For instance, a study conducted by Wang and colleagues investigated 974 critically ill children between 1 month and 18 years with sepsis. The researchers assessed the risk of inpatient mortality based on serum magnesium levels at admission, and found that the hypermagnesemia group had a 6-fold higher in-hospital mortality rate than the normal group (14.5% vs. 2.4%, *p* < 0.001) [54].

In conclusion, hypomagnesemia is associated with an increased risk of infections, worsened sepsis progression, and decreased survival rates in critically ill patients. Therefore, it is advisable to consider the use of magnesium supplementation, aiming at serum magnesium levels within the upper limit of the normal range (2.1–2.3 mg/dL). This approach can be employed both as a preventive and therapeutic measure, in conjunction with standard medical practices, particularly in elderly patients, those with chronic kidney disease, and individuals with heart failure. However, due to conflicting results in the pediatric population, this recommendation is limited to critically ill adults.

**Table 1 vaccines-11-01122-t001:** Summary of the main studies investigating the role of magnesium in sepsis and septic shock.

Study (Year), Study Design	Field of Study	Population	Intervention	Findings
Tonai (2022) [53], Observational retrospective	Hypomagnesemia and coagulopathy in septic patients	753 patients	No intervention	Hypomagnesemia independently associated with risk of DIC (OR 1.69)
Wang (2022) [54], Observational retrospective	Mg levels: mortality predictor in critically ill children with sepsis	974 patients	No intervention	Hypermagnesemia associated with a six-fold increase in mortality; AKI and liver dysfunction higher in both hypermagnesemia and hypomagnesemia, compared to normal concentrations
Noormandi (2019) [51], RCT	Mg on lactate clearance	58 patients (30 Mg group; 28 placebo group)	Mg administration aiming > 2 mg/dL for 3 days	Mg supplementation increased lactate clearance, and reduced both the time to lactate clearance and the ICU LOS
Thongprayoon (2015) [46], Observational retrospective	Mg levels and septic shock	2589 patients with SIRS	No intervention	Hypomagnesemia associated with increased risk of developing septic shock (OR 1.86). Mg 2.1–2.3 mg/dL has the lowest septic shock incidence
Huang (2015) [55], Observational retrospective	Association between dietary or plasma Mg and diabetes incidence, and with mortality, in the elderly	1400 patients aged ≥ 65 years	No intervention	Normal and high plasma Mg in conjunction with high DDS had relative risks of 0.58 and 0.46 in mortality, compared to low plasma Mg and lower DDS

Abbreviations: AKI, acute kidney injury; DIC, disseminated intravascular coagulation; DDS, dietary diversity score; ICU, intensive care unit; LOS, length of stay; Mg, magnesium; RCT, randomized controlled trial; SIRS, systemic inflammatory response syndrome.

### 3.3. Respiratory Diseases

Magnesium may induce bronchodilation and reduce inflammation [34,57], by influencing calcium dynamics via a blockage of the voltage-dependent calcium channels [58,59,60,61]. Moreover, this ion plays a crucial role in reducing airway hyper-reactivity and wheezing [57,62], and has the potential to prevent pulmonary fibrosis by reducing TGFb1 release, and subsequent intrapulmonary collagen deposition [63].

It must be noted that hypermagnesemia can lead to severe symptoms, such as flaccid muscle paralysis, hyporeflexia, bradycardia, respiratory depression, coma, and cardiac arrest [64,65,66]. Conversely, hypomagnesemia has been associated with a number of clinical manifestations, including respiratory muscle weakness and bronchospasm [67,68,69,70].

Several studies have investigated the importance of magnesium homeostasis in respiratory diseases.

Thogprayoon et al. [71] analyzed a group of 9780 patients with acute respiratory failure (ARF) not requiring mechanical ventilation on admission. They reported the lowest incidence of ARF when the serum magnesium level on admission was within 1.7–1.9 mg/dL. Moreover, the risk of developing in-hospital ARF requiring mechanical ventilation was increased with both hypomagnesemia (<1.7 mg/dL) and hypermagnesemia (>1.9 mg/dL) at the time of admission. However, the effects of the correction of magnesium levels, and its impact on the risk of ARF requiring mechanical ventilation, were not available. A single-center retrospective study [72] investigated the relationship between serum magnesium levels in the first 48 h, and the 30-day mortality amongst patients admitted for community-acquired pneumonia (CAP). Interestingly, the study showed a U-shaped relationship, with the lowest incidence of 30-day mortality occurring when serum magnesium was within the range of 1.35–2 mg/dL. The study also found that patients with borderline-elevated magnesemia (2–2.4 mg/dL) had higher 30-day mortality, compared to those with borderline-low magnesemia (1.35–2 mg/dL). Again, no data on the normalization of magnesium levels were provided. Another study, by Broman et al. [73], rejected the hypothesis that magnesium supplementation in patients with CAP could improve outcomes, as mild hypermagnesemia was associated with markedly worse survival compared to normal magnesemia. Cirik et al. [74] studied the effect of admission magnesium levels on the length of stay in ICU, length of mechanical ventilation, and 28-day mortality in patients admitted to the ICU for ARF. The study demonstrated that hypomagnesemia and normomagnesemia did not affect these outcomes, while hypermagnesemia was associated with significantly higher mortality rates. It is important to note that hypomagnesemia was routinely corrected through magnesium supplementation upon diagnosis, but the investigators did not conduct repeated measurements of serum magnesium levels.

Based on these findings, it is worth noting that certain reports have highlighted a correlation between severe COVID-19 symptoms and low serum levels of magnesium [75]. This observation has led some authors to hypothesize that magnesium deficiency could potentially exacerbate the inflammation induced by SARS-CoV-2, and contribute to the progression of the disease [76].

In a recent cohort observational study [77], it was found that magnesium, vitamin C, and vitamin D supplementation in COVID-19 patients over 50 years old was associated with a significant reduction in oxygen supplementation and intensive care support. The study reported no deaths in either group during the follow-up period. However, this paper reported several limitations, including a small sample size (43 patients, versus the 56 suggested by a post hoc estimation of the necessary sample size), a control group with older patients afflicted by multiple comorbidities and, lastly, magnesium levels not being measured at admission or during the follow-up process.

Although the exact role of magnesium in the treatment of asthmatic status is not completely understood, its supplementation is part of current clinical practice as a last resort for severe and persistent asthma [13,78,79,80]

In addition to its inhibitory effect on calcium-mediated smooth muscle contraction, magnesium is involved in preventing mast cell degranulation, by inhibiting the production of oxygen free radicals [81], and magnesium supplementation might downregulate neutrophil respiratory burst in asthmatic patients, by increasing the level of cyclic adenosine monophosphate (cAMP) [82].

Despite several trials showing that magnesium plays a role in managing acute asthmatic exacerbations, the protective effect of magnesium is controversial, as shown by Bokhari and colleagues [83]. It should be noted that only two studies reported data on mortality. A nationwide retrospective study in Japan found no significant association between intravenous MgSO_4_ use and mortality in patients with severe acute asthma [84]. The study had limitations, including the rarity of asthma-related mortality, and the lack of standardized magnesium supplementation doses.

In a double-blind, placebo-controlled trial by Goodacre et al. [85], IV or nebulized magnesium sulfate did not significantly reduce hospital admission rates, or improve breathlessness symptoms, in adults with severe acute asthma. However, the study was underpowered to detect differences in mortality due to exclusion criteria.

In conclusion, there is insufficient evidence to determine the impact of magnesium on mortality in acute asthmatic exacerbation. Analyzing the mortality in this population poses a significant challenge, as it is an infrequent outcome, and thus can influence the reliability and interpretation of study findings. A summary of the evidence is provided in Table 2.

### 3.4. Cardiac Surgery

We have previously reported the relationship between magnesium deficiency and the elevation of inflammatory mediators, specifically CRP [86]. After a heart attack, CRP activates the complement system on ischemic myocytes, leading to their lysis. Hence, serum magnesium levels are hypothesized to influence the development of heart disease [87]. Furthermore, it has been observed that a low serum magnesium concentration induces endothelial dysfunction, through the NF-κB signaling pathway [88], and atherosclerosis [35,56,89], and increases platelet aggregation [90]. Notably, a recent meta-analysis showed a significant inverse association between dietary magnesium intake (resulting in decreased serum concentration) and the overall risk of cardiovascular events [91]. It is important to consider these findings in the context of cardiovascular health, and the potential implications of magnesium deficiency.

Several studies have investigated the occurrence of hypomagnesemia after cardiac surgery. Hypomagnesemia is a common complication after cardiac surgery, and has been associated with an increased risk of major adverse cardiac events [92,93,94,95,96,97,98]. In addition, low serum magnesium levels in ICU patients have been linked to prolonged mechanical ventilatory support, higher incidence of rhythm disorders, and increased mortality rates [99,100]. Several randomized controlled trials (RCTs) have investigated the effect of magnesium supplementation on the acute phase of myocardial infarction, but the results have been conflicting [101,102,103]. Likewise, the efficacy of magnesium supplementation in preventing atrial fibrillation following coronary artery bypass grafting (CABG) remains controversial, as reported in multiple trials and meta-analyses [104,105,106,107,108,109,110,111,112,113,114,115,116,117]. To further explore the potential benefits of magnesium supplementation in CABG patients, Carrió et al. conducted an RCT. However, consistent with previous studies, the results did not demonstrate a favorable effect on clinical outcomes [118].

### 3.5. Neurological Intensive Care

Few studies have addressed serum magnesium levels and clinical outcomes in the neurointensive care unit. Wang et al. [119] investigated the association between initial serum magnesium levels and mortality in traumatic brain injury patients (TBI). The study included 2280 adult patients with an abbreviated injury score greater than 3. Serum magnesium ranging from 1.7 to 2.4 mg/dL was considered normal. The authors reported that both hyper- and hypomagnesemia were associated with higher mortality rates. After adjusting for multiple possible confounders, including the Glasgow Coma Scale (GCS) score, a higher serum magnesium level was still associated with 30-day mortality. Furthermore, stepwise multivariate logistic regression also confirmed the positive association between serum magnesium and mortality after including GCS. The Spearman correlation test suggested that a higher GSC was associated with lower Mg (i.e., the higher mortality of patients with lower serum magnesium levels might be mainly due to more severe brain injury, rather than the independent effect of magnesium on the injured central nervous system). The authors concluded that TBI induces an acute inflammatory state that increases brain metabolism, which could consume the stored magnesium. Gastrointestinal loss, kidney dysfunction, and diuretic use could all contribute to hypomagnesemia, while acidosis and severe trauma promote hypermagnesemia. The limitations of this study were the lack of cerebrospinal fluid magnesium dosing, the use of ionized magnesium that did not represent the body’s total magnesium storage, and magnesium dosing being done only one time, at admission.

Ardehali et al. [120] investigated the association of serum magnesium with neurosurgical ICU clinical outcomes. The study included 210 adult postoperative patients with normal magnesium, ranging from 1.7 to 2.1 mg/dL. The authors found no correlation between serum magnesium levels at admission and mortality, length of stay, duration of mechanical ventilation, or sofa score (Table 3).

Overall, these studies suggest that the relationship between serum magnesium levels and clinical outcomes in the neuro ICU is complex, and warrants further investigation.

### 3.6. Kidney Injury and Electrolyte Disorders

Magnesium is a crucial element in maintaining renal function, as it preserves kidney function when administered during nephrotoxic acute kidney injury (AKI) [121]. Magnesium enhances kidney blood flow via an endothelium-dependent release of NO [122], counteracting the vasoconstriction induced by endogenous catecholamines, and favoring vasodilation [123]. Therefore, hypomagnesemia may increase the risk of AKI, by disrupting the kidneys’ vascular autoregulation.

A prospective multicenter observational study by Ribeiro et al. [124], involving 7042 critically ill patients, demonstrated that both hypomagnesemia and hypermagnesemia were associated with higher mortality and renal dysfunction (65% vs. 52%, respectively). Specifically, hypomagnesemic patients were 25% more likely to develop AKI, and require dialysis and a longer hospital stay, strongly supporting the belief in the protective role of magnesium on kidney function. In contrast, hypermagnesemia did not show a significant association with AKI onset or lower non-recovery rates, defined as maintenance of the same or a higher AKI stage [125]. These outcomes have been consistently demonstrated by Koh et al. [126], in a retrospective observational cohort study on 9766 patients after cardiac surgery. The study showed a progressively increased prevalence of AKI with lower magnesium levels. Furthermore, it confirmed the association between preoperative hypomagnesemia and an elevated risk of developing AKI and AKI requiring dialysis (OR 1.39 and 1.67, respectively). It has also been suggested that magnesium supplementation may play a protective role in AKI induced by vancomycin plus piperacillin-tazobactam, reducing the incidence of AKI by 20% [127].

Detrimental effects of hypomagnesemia have also been reported in kidney transplant patients, linking low serum magnesium levels to a decline in kidney allograft function, and an increased rate of loss of kidney graft [128].

In contrast to these findings, Isakov et al. [129] reported a beneficial effect of hypomagnesemia in kidney transplant patients. Their study found that for every 0.1 mg/dL increase in serum magnesium, the risk of death or graft loss increased by 10%. As a result, hypomagnesemia was associated with better patient and allograft survival for up to 10 years post-transplant. In addition, a double-blind randomized study that included 54 hemodialysis patients showed that magnesium supplementation did not improve endothelial function [130]. A possible explanation for this result is that the impact of endotoxin accumulation in chronic kidney disease may outweigh the potential benefits of magnesium supplementation on endothelial function.

Contradictory findings have also been observed when studying the impact of abnormal magnesium levels on mortality and AKI onset in critically ill children. A retrospective study [130] involving 3669 children found that both hypermagnesemia and hypomagnesemia were independently associated with 28-day mortality. However, in the multivariable analysis, only hypermagnesemia was associated with AKI onset, and no association was reported between serum magnesium levels and AKI stages.

In conclusion, conflicting data exist regarding the protective role of magnesium in kidney function. Most of the evidence supports a strong protective role of magnesium in the prevention of AKI and AKI requiring dialysis in critically ill adults, with a potential role in promoting recovery after AKI. However, the data regarding kidney transplant patients and the pediatric population are currently unclear, and warrant further research. Additionally, no study has found an association between the trend of magnesium concentration and kidney injury in critically ill patients, except in the context of kidney transplantation, as mentioned above. A summary of the evidence is provided in Table 4.

## 4. Conclusions

The overall evidence provided in this review emphasizes the importance of maintaining magnesium homeostasis in critical care. The interplay between magnesium and the immune system has significant clinical implications. Hypomagnesemia has been associated with an increased risk of infections and sepsis, and decreased survival rates, in critically ill patients. In respiratory diseases, it has been linked to respiratory muscle weakness and bronchospasm. On the other hand, hypermagnesemia can cause flaccid muscle paralysis, hyporeflexia, bradycardia, respiratory depression, coma, and cardiac arrest. The impact of correcting magnesium levels on the risk of developing acute respiratory failure requiring mechanical ventilation remains unknown, and warrants further investigation. Furthermore, magnesium plays an important role in protecting the kidney from AKI, and reducing the risk of progression to AKI requiring dialysis. The current literature suggests there are benefits of maintaining normal magnesemia in critically ill patients, such as reducing overall mortality, modulating the inflammatory response, and protecting the kidney. Overall, the findings emphasize the importance of implementing strategies for monitoring and managing magnesium levels in critical care, to optimize patient outcomes.

## Figures and Tables

**Table 2 vaccines-11-01122-t002:** Summary of studies investigating the role of magnesium in respiratory diseases in critical care settings.

Study (Year), Study Design	Field of Study	Population	Intervention	Findings
Tan (2020) [77], Observational prospective	Effect of combination of Mg, Vit D, and Vit B12 on progression to severe COVID-19	43 (17 treatment group; 26 control group)	150 mg/d oral Mg + 100 IU/d oral Vit D3 + 500 mcg/d oral Vit B12	No deaths in either group
Cirik (2020) [74], Observational retrospective	Effect of admission Mg serum levels in ARF on ICU LOS, length of MV, and 28-day mortality	329 patients	No intervention	ICU mortality significantly higher in the hypermagnesemia group than the other groups. No significant effect on length of hospital stays, ICU LOS, length of MV, and 28-day mortality
Nasser (2018) [72], Observational retrospective	Effect of Mg serum levels in the first 48 h on 30-day mortality in hospitalized CAP	3851 patients with CAP	No intervention	Hypomagnesemia and hypermagnesemia on admission associated with an increased rate of 30-day mortality
Thongprayoon (2015) [71], Observational retrospective	Correlation of Mg levels in the first 24 h and in-hospital ARF requiring mechanical ventilation	9780 patients	No intervention	Admission hypomagnesemia and hypermagnesemia associated with an increased risk of in-hospital ARF requiring mechanical ventilation
Hirashima (2016) [84], Observational retrospective	Effect of IV Mg supplementation in severe asthma exacerbations	599 pairs matched with propensity score	Any doses of IV MgSO_4_ within 2 days of admission	No significant benefit of IV MgSO_4_ on 7-, 14- and 28-day mortalities
Goodacre (2013) [85], RCT	Effect of IV Mg and NEB Mg supplementation in severe asthma exacerbations	1087 patients (396 IV Mg group; 333 NEB Mg group; 358 placebo group)	2 g IV MgSO_4_ or 1.5 g NEB MgSO_4_ or placebo over conventional therapy	Mg did not significantly reduce hospital admission rates or improve breathlessness symptoms; not enough power to detect mortality differences

Abbreviations: ARF, acute respiratory failure; CAP, community-acquired pneumonia; ICU, intensive care unit; LOS, length of stay; Mg, magnesium; MgSO_4_, magnesium sulfate; MV, mechanical ventilation; IV, intravenous, NEB, nebulized.

**Table 3 vaccines-11-01122-t003:** Summary of studies investigating the role of magnesium in the neuro intensive care unit.

Study (year), Study Design	Field of Study	Population	Notes	Findings
Wang (2022) [119], Observational retrospective	Effect of admission Mg serum levels on mortality in TBI	2280 patients with TBI	Eumagnesemia defined as: 1.7–2.4 mg/dL	TBI patients with lower and higher serum Mg levels had higher mortality rates (lowest mortality in patients with Mg 1.7–2 mg/dL)
Ardehali (2017) [120], Observational prospective	Effect of admission Mg serum levels in neurosurgical ICU	210 patients	Eumagnesemia defined as: 1.7–2.1 mg/dL	No relation between admission serum Mg level and mortality

Abbreviations: ICU, intensive care unit; Mg, magnesium; TBI, traumatic brain injury.

**Table 4 vaccines-11-01122-t004:** Summary of studies investigating the relationship between magnesium, kidney injury, and electrolyte disorders in critical care.

Study (Year), Study Design	Field of Study	Population	Intervention	Findings
Isakov (2022) [128], Observational retrospective	Effects of hypomagnesemia on post-transplantation survival and graft loss	726 patients	Serum levels measured from one month to one year post-transplant	Post-transplant hypomagnesemia (<1.7 mg/dL) independently associated with better patient and allograft survival
Koh (2022) [126], Observational retrospective	Preoperative ionized Mg serum levels and risk of AKI (cardiac surgery)	9766 patients	Mg levels measured before surgery	OR for postoperative AKI progressively larger with progressively lower serum Mg concentration. Serum Mg level < 1.09 mg/dL associated with AKI and AKI requiring dialysis
Ribeiro (2022) [124], Observational prospective	Effect of admission Mg serum levels on kidney and mortality outcomes	7042 patients:	Mg serum levels measured at ICU admission	Hypomagnesemia associated with 25% increased risk of AKI, 65% increased mortality, and longer hospital stay. Hypermagnesemia associated with lower kidney non recovery rates and 52% increased mortality risk
Morooka (2021) [131], Observational retrospective	Impact of Mg serum levels on death and AKI onset	3669 children	No intervention	Both hypo and hypermagnesemia associated with 28-day mortality (OR 2.99 and 2.80, respectively). Hypermagnesemia associated with AKI (OR 1.52). No association with AKI stage.
Khalili (2021) [127], RCT	IV Mg for prevention of VPT-induced AKI in critically ill patients	30 patients	No intervention	Administration of Mg with target serum level around 3 mg/dL associated with reduced incidence of AKI in critically ill patients receiving VPT
Mortazavi (2013) [130], RCT	Effect of Mg supplementation on endothelial function	54 hemodialysis patients	Oral Mg supplementation (440 mg 3 times per week for 6 months)	No association between Mg supplementation and endothelial function

Abbreviations: AKI, acute kidney injury; ICU, intensive care unit; IV, intravenous; Mg, magnesium; OR, odds ratio; VPT, vancomycin plus piperacillin-tazobactam.

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
