# Peer review of "Role of Magnesium in the Intensive Care Unit and Immunomodulation: A Literature Review"

_vaccines, 2023, doi:10.3390/vaccines11061122_

Round 1

Reviewer 1 Report

The manuscript reviews the literature on the role of magnesium in intensive care units and immunomodulation”, using PubMed, Central, and the US Nat Inst of Healthy Clinical Trials using 6 keywords. Addresses the possible deregulation of the immune response induced by the electrolyte in the homeostasis of individuals with sepsis submitted to an intensive care environment and its connection with mortality in the ICU.

-The topic addressed is interesting and important and deserves consideration in the area of study. However, in the description, we do not feel an improvement in the subject that would differentiate it from other previous works. The work was limited to describing the number of cases, not delving into the mechanism of action or pathogenicity at the molecular level of the lack of electrolytes.

-In MM, the authors inform that 1083 papers were selected and duplicates and others that were not suitable were excluded, but they still need to report how many articles were computed at the end of the study.

-This point is related to the number of examples presented in tables 1, 2, 3, and 4, which ranged from 2 to 4, although some of these citations performed meta-analysis.

-The conclusions of the authors, although adequate to what they proposed, seemed to me to be very superficial, adding very little to the knowledge of the subject

-Both the abstract and the introduction must be better presented, and each reference must be described in a certain way.

Minor editing of English language required.

Author Response

The manuscript reviews the literature on the role of magnesium in intensive care units and immunomodulation”, using PubMed, Central, and the US Nat Inst of Healthy Clinical Trials using 6 keywords. Addresses the possible deregulation of the immune response induced by the electrolyte in the homeostasis of individuals with sepsis submitted to an intensive care environment and its connection with mortality in the ICU.

-The topic addressed is interesting and important and deserves consideration in the area of study. However, in the description, we do not feel an improvement in the subject that would differentiate it from other previous works. The work was limited to describing the number of cases, not delving into the mechanism of action or pathogenicity at the molecular level of the lack of electrolytes.

We would like to thank the Reviewer for the precious comments. We have extensively revised the overall manuscript and added important parts to the introduction and to all the sections. Specifically, we added a new chapter (3.1) talking about the interaction between Mg and the immune system at a molecular level. In the introduction we gain some insight into the role of Mg in the ICU and we have better explained the aim of this review: to understand how the interaction between Mg and immune system may affect mortality in diseases that are seen in critically ill patients (there are no other reviews in the literature looking at this, although Mg is used frequently by anesthesiologists). Given the reviewer comments we added specific paragraphs at the beginning of the different chapters, linking the hypothesized pathogenic and molecular mechanisms to the clinical outcomes described.

-In MM, the authors inform that 1083 papers were selected and duplicates and others that were not suitable were excluded, but they still need to report how many articles were computed at the end of the study.

We added the required information at the end of the Methods section.

-This point is related to the number of examples presented in tables 1, 2, 3, and 4, which ranged from 2 to 4, although some of these citations performed meta-analysis.

We have updated and edited all the tables, making them more clear. As previously stated, our focus was on papers reporting mortality outcomes.

-The conclusions of the authors, although adequate to what they proposed, seemed to me to be very superficial, adding very little to the knowledge of the subject

We thank the Reviewer for the useful advice. We have now expanded the conclusion.

-Both the abstract and the introduction must be better presented, and each reference must be described in a certain way.

We have updated the introduction and abstract as requested. We also synthesized some of the references, to make the flow of reading smoother. Thanks!

Reviewer 2 Report

The review titled “Role of magnesium in intensive care unit and immunomodulation: a review” by Saglietti, E. et al is very well written.  Some minor corrections are listed below:

Line 80-81 : “Noormandi et al..” please indicate the reference no. here.

Line 96 :  “ Wang and colleagues…” please indicate the reference no. here.

Line 105-106:  “On the other hand, hy-104 permagnesemia can induce bronchodilation and immunomodulation by reducing inflammation. As such, it could prove to be useful in several respiratory diseases.”  Reference is required for this statement. 

Line 109:  “ Thogprayoon et al.…” please indicate the reference no. here.

Line 110-111:  Statement “They reported the lowest incidence of 110 ARF when admission serum magnesium was within 1.7-1.9 mg/dl”.  Is “admission serum” referring to serum levels at admission of patient? Please rephrase accordingly.

Line 162-163:  Statement “Wang et al (74) investigated the association between initial serum magnesium levels and mortality in traumatic brain injury patients (TBI).”  Is the term “initial Serum” referring to serum levels of patient at the time of admission?  Please rephrase accordingly.

Author Response

The review titled “Role of magnesium in intensive care unit and immunomodulation: a review” by Saglietti, E. et al is very well written.  Some minor corrections are listed below:

We thank the Reviewer for the contribution.

Line 80-81 : “Noormandi et al..” please indicate the reference no. here.

We have fixed the reference.

Line 96 :  “ Wang and colleagues…” please indicate the reference no. here.

Line 105-106:  “On the other hand, hy-104 permagnesemia can induce bronchodilation and immunomodulation by reducing inflammation. As such, it could prove to be useful in several respiratory diseases.”  Reference is required for this statement. 

We have provided the reference in the revised version of the manuscript.

Line 109:  “ Thogprayoon et al.…” please indicate the reference no. here.

We have fixed the reference.

Line 110-111:  Statement “They reported the lowest incidence of 110 ARF when admission serum magnesium was within 1.7-1.9 mg/dl”.  Is “admission serum” referring to serum levels at admission of patient? Please rephrase accordingly.

Yes. We have changed and rephrased that part.

Line 162-163:  Statement “Wang et al (74) investigated the association between initial serum magnesium levels and mortality in traumatic brain injury patients (TBI).”  Is the term “initial Serum” referring to serum levels of patient at the time of admission?  Please rephrase accordingly.

We have rephrased the sentence, thanks.

Reviewer 3 Report

This review manuscript discuss about Role of magnesium in intensive care unit and immunomodulation: a review"

An interesting knowledge has been reported. however the following comments should be addressed before acceptance

Major revision

1. Novelty of the manuscript must be better emphasized

2.  the importance of the work should be mentioned in the abstract section

3.  author should add some more discussion points about the significance of this review paper in introduction section

4.  add main role of Sepsis and septic shock 

5. how does concentration (mg) effects the kidney injuries

6.  suggested to add some recent litterature survey (references). current form author used old reference

7. there are some typological errors are present in the manuscript that should be revised carefully

Moderate correction required

Author Response

This review manuscript discuss about Role of magnesium in intensive care unit and immunomodulation: a review"

An interesting knowledge has been reported. however the following comments should be addressed before acceptance

Major revision

  1. Novelty of the manuscript must be better emphasized

We thank the Reviewer for the relevant comments. We have now emphasized the importance of this review in the introduction (which was also expanded) and substantially revised the overall manuscript, adding new parts to all the sections. 

  1. the importance of the work should be mentioned in the abstract section

We have edited the abstract in accordance.

  1. author should add some more discussion points about the significance of this review paper in introduction section

Thanks, we improved the introduction.

  1. add main role of Sepsis and septic shock 

We have updated the section on Sepsis.

  1. how does concentration (mg) effects the kidney injuries

At the moment there is unclear data in kidney transplanted population and pediatric population, further research is needed. Moreover no study found any association between magnesium concentration trend and kidney injury in critically ill patients, it has only been found in transplanted patients where Isakov et al found that every increase in Mg of 0.1 mg/dL, augmented the risk for death or graft loss was 10% higher. We have updated the section on renal disease in accordance.

  1. suggested to add some recent litterature survey (references). current form author used old reference

We have added some recent references, staying in the scope of our analysis, and edited the manuscript accordingly.

  1. there are some typological errors are present in the manuscript that should be revised carefully

We have revised the manuscript and made the requested changes.

Round 2

Reviewer 1 Report

The authors conducted An extensive review, incorporating most of the suggestions and criticisms. We, therefore, now consider that the manuscript may be accepted for publication,

No major comments.

Reviewer 3 Report

Accept

Authors addressed all the comments and thus improved the quality of this manuscript